# Immunological and Oxidative Biomarkers in Bovine Serum from Healthy, Clinical, and Sub-Clinical Mastitis Caused by *Escherichia coli* and *Staphylococcus aureus* Infection

**DOI:** 10.3390/ani13050892

**Published:** 2023-03-01

**Authors:** Asmaa Sadat, Alshimaa M. M. Farag, Driss Elhanafi, Amal Awad, Ehab Kotb Elmahallawy, Noorah Alsowayeh, Manal F. El-khadragy, Gehad E. Elshopakey

**Affiliations:** 1Department of Bacteriology, Mycology, and Immunology, Faculty of Veterinary Medicine, Mansoura University, Mansoura 35516, Egypt; 2Department of Internal Medicine and Infectious Diseases, Faculty of Veterinary Medicine, Mansoura University, Mansoura 35516, Egypt; 3Biomanufacturing Training and Education Center, North Carolina State University, Raleigh, NC 27606, USA; 4Department of Zoonoses, Faculty of Veterinary Medicine, Sohag University, Sohag 82524, Egypt; 5Department of Biology, College of Education (Majmaah), Majmaah University, Al-Majmaah 11952, Saudi Arabia; 6Department of Biology, College of Science, Princess Nourah bint Abdulrahman University, P.O. Box 84428, Riyadh 11671, Saudi Arabia; 7Department of Clinical Pathology, Faculty of Veterinary Medicine, Mansoura University, Mansoura 35516, Egypt

**Keywords:** mastitis, *E. coli*, *S. aureus*, oxidative/antioxidant molecules, APP, inflammatory cytokines

## Abstract

**Simple Summary:**

Establishing reliable biomarkers of udder bacterial infection and its bovine immune response at the early stage of the mastitic infection is considered an urgent matter. Thus, reliable biomarkers were measured in our study for accurate detection of the changes in biochemical and immunological parameters related to both clinical and sub-clinical mastitis. Ultimately, APP and cytokines, along with antioxidant markers can be used as early indicators of subclinical and clinical mastitis.

**Abstract:**

The study aimed to investigate the mastitis’ emerging causative agents and their antimicrobial sensitivity, in addition to the hematological, biochemical indicators, oxidative biomarkers, acute phase protein (APP), and inflammatory cytokine changes in dairy farms in Gamasa, Dakahlia Governorate, Egypt. One hundred Holstein Friesian dairy cattle with clinical and subclinical mastitis were investigated and were allocated into three groups based on a thorough clinical examination. *Escherichia coli* and *Staphylococcus aureus* were found responsible for the clinical and subclinical mastitis in dairy farms, respectively. Multiple drug resistance (MDR) was detected in 100%, and 94.74% of *E. coli* and *S. aureus* isolates, respectively. Significantly low RBCs count, Hb, and PCV values were detected in mastitic cows compared with both subclinical mastitic and control groups; moreover, WBCs, lymphocytes, and neutrophil counts were significantly diminished in mastitic cows compared to the controls. Significantly higher levels of AST, LDH, total protein, and globulin were noticed in both mastitic and subclinical mastitic cows. The haptoglobin, fibrinogen, amyloid A, ceruloplasmin, TNF-α, IL-1β, and IL-6 levels were statistically increased in mastitic cows compared to the controls. Higher MDA levels and reduction of TAC and catalase were identified in all the mastitic cases compared to the controls. Overall, the findings suggested potential public health hazards due to antimicrobial resistance emergence. Meanwhile, the APP and cytokines, along with antioxidant markers can be used as early indicators of mastitis.

## 1. Introduction

In veterinary medicine, bovine mastitis is considered one of the most common and economically important diseases affecting dairy herds worldwide. It mostly causes major economic losses [1,2]. Hence, the economic impact of mastitis is usually due to decreased milk production, increased costs of veterinary treatment, and premature culling of infected animals [3]; in addition, it causes alterations in the milk and udder physically, chemically, and pathologically [4,5].

*Staphylococcus aureus* is one of the major common pathogens causing mastitis in dairy animals [6]. The ability of the organism to cause infections is probably due to the expression of various toxins, virulence factors, and cell wall adhesion proteins. The bacterium can survive phagocytosis in the udder and often causes chronic inflammation [7]. These infections frequently do not respond to routine therapy. However, other microorganisms may be attributed to mastitic infections like *Escherichia coli, Salmonella* spp., and *Streptococci* or *Listeria* spp. [8,9,10].

Bovine mastitis poses a great one-health concern and zoonotic danger to humans due to the capability of this food-borne pathogen to be transmitted through milk or even through workers to consumers [3] The use of antimicrobial agents on cattle farms is often useful but the excessive use of it to control or prevent the mastitic infections may be contributing to the global antimicrobial resistance development [3]. Besides, antimicrobial residue leads to a decrease in the quality of milk [11]. Bovine mastitis is classified into two categories: clinical mastitis and subclinical mastitis; clinical mastitis is characterized by observable clinical signs of the udder and signs of inflammation with general health disturbance [12,13,14,15]; subclinical mastitis is difficult to be detected at once due to the lack of visible signs. It takes days or weeks to be observed [13], producing huge losses in the milk production due to the long-time persistence of infection and alterations of physical and chemical properties of the milk [16]. Usually, subclinical mastitis is more common than clinical mastitis [17,18].

The number of Somatic cells in the milk may be considered a diagnostic marker for subclinical mastitis [13]; these somatic cells consist of macrophages, neutrophils, lymphocytes, and some mammary epithelial cells. In the case of the healthy udder, macrophages are the predominant cell, while during early mastitis neutrophils become the predominant cells [19]. Host-pathogen interactions lead to the activation of the innate immune response to recognize infected microorganisms or rechange immune components by increasing the number of macrophages and releasing cytokines, which recruit leukocytes at the site of infection and trigger the local and systemic acute phase response [19,20]. Furthermore, acute phase proteins such as haptoglobin, cathelicidins, and peptidoglycan recognition protein were recorded at higher rates during infection [21]; similarly, high levels of IL-1β, IL-8, and TNF-α were expressed [22].

The pathogenesis of udder bacterial infection and bovine immune response must be studied to establish reliable biomarkers which can be detected at the early stage of the infection. Thus, these reliable biomarkers were measured in our study for accurate detection of the changes assigned with the bacteriological agent invasion and accompanied changes in biochemical and immunological parameters related to both clinical and sub-clinical mastitis. Meanwhile, the antimicrobial resistance of our strains was assessed to measure the current danger of multidrug resistance strains affecting public health.

## 2. Materials and Methods

### 2.1. Animal Population

The present study was conducted in January 2021 on 100 dairy cows: healthy cows (*n* = 20), clinical mastitis-infected cows (*n* = 30), and subclinical mastitis-infected cows (*n* = 50) from two different farms located on the road of Gamasa city, Dakahlia Governorate, Egypt. Their ages ranged between 4 and 7 years. All dairy cattle diagnosed in our study were Holstein Friesian. The study was performed on cows that had not received any medication in the week before sampling. On the farm, modern management techniques and good hygiene standards were applied. Only automatic milking machines were used at both farms.

All investigated cows were subjected to a thorough clinical examination. After milking the diseased cows, the size, the conformation of the udder, and the relative size of all quarters were inspected. The udder tissue and supramammary lymph nodes were also thoroughly examined. Cows with clinical mastitis were identified if one or more of the following signs were observed: cardinal signs of inflammation in one or more of an udder’s quarters, signs of systemic reactions (fever, depression, and disturbed appetite), or abnormal physical character of milk (clot formation, discoloration, and alterations in viscosity, aberrant smell, and the presence of blood).

Due to the absence of observable clinical signs in cows with subclinical mastitis, the presumptive diagnosis was based on the laboratory diagnostic tests of milk samples, the California mastitis test (CMT), and somatic cell count (SCC). Cows with positive CMT or having SCC > 200,000 cells/mL and lacking clinical signs were considered affected with subclinical mastitis.

Briefly, the California mastitis test (CMT), [23] was performed as follows: a plastic vessel with 4 shallow wells was used for collecting approximately 2 mL of milk from each udder quarter. Then, an equal amount of alkali reagent (kerbl^®^ reagent) was added. A gentle circular motion was applied to the mixtures in a horizontal plane for 5 s and the different degrees of gel were recorded, according to the system used in the Nordic countries as the scoring is made from 1–5. Meanwhile, all the milk samples were examined automatically for somatic cell count by using The Nucleo Counter^®^ SCC-100™. The sample was warmed in a water bath at 35 °C for 5 min and then mixed automatically before reading [24].

All procedures were performed following the principles and specific guidelines presented in the Mansoura University Animal care and approved by its Ethical Committee.

### 2.2. Sample Collection

Milk and blood samples were collected from each individual dairy cow (*n* = 100), including healthy cows (*n* = 20), clinical (*n* = 30), and subclinical mastitis cows (*n* = 50). The teats were wiped using alcohol for sterilization and all samples were taken in sterilized vials. After milk samples were taken aseptically, they were transported in coolers (4 to 8 °C) for subsequent bacteriological analyses within 6 hrs. All milk samples were sent to the Department of Bacteriology, Mycology and Immunology lab, Faculty of Veterinary Medicine, Mansoura University, Egypt.

For blood samples, the halter was used to position the animal’s head in a slightly elevated manner to expose the jugular vein with minimal restraint to get the blood samples without causing injury. One blood sample was collected with an anticoagulant (EDTA) for complete blood counting. The second blood sample was collected in a heparinized tube which was rapidly centrifuged at 3000× *g* for 10 min for separation of blood plasma. The collected plasma was used for the estimation of fibrinogen. The third blood sample was collected in plain test tubes, left for 15 min to clot, and centrifuged at 3000 rpm (4 °C) for serum separation. The separated serum was stored at −80 °C for further determination of biochemical, inflammatory, and oxidative stress/antioxidant parameters.

### 2.3. Bacterial Identification

All collected milk samples were centrifuged for 10 min. A drop from the sediment was cultivated onto Baird-Parker agar, 5% sheep blood agar, MacConkey’s agar, and Mannitol salt agar, (Oxoid, Ltd., Basingstoke, UK) at 37 °C for 24–48 h. Bacterial colonies were classified according to their phenotypic characters on the culture media. All milk samples that give more than two bacterial species on bacterial culture plates were considered contaminated samples and were discarded. Presumptive characterization of the isolated bacteria was carried out based on Gram’s stain and biochemical characteristics according to Quinn and his group [25].

### 2.4. Molecular Confirmation of S. aureus and E. coli Isolates

#### 2.4.1. DNA Extraction

The DNA was extracted from all suspected isolates by the boiling method [26]. Two to three colonies from suspected *S. aureus* and *E. coli* isolates were suspended in 200 uL deionized free water and boiled for 10 min, and then centrifugated for 10 min. The supernatants were transferred to a new sterile Eppendorf tube and stored at −20 °C as a DNA sample.

#### 2.4.2. Molecular Confirmation Using Polymerase Chain Reaction (PCR) for *S. aureus* and *E. coli* Isolates

All the isolates suspected to be *S. aureus* by biochemical typing were confirmed by amplification of species-specific *nuc* gene (encoding for the *S. aureus*-specific thermonuclease) using the primers: *nuc*-F:(GCGATTGATGGTGATACGGTT) and *nuc*-R: (AGCCAAGCCTTGACGAACTAA AGC) by using PCR amplification technique according to Sallam et al. [27]. In brief, A 25 μL PCR reaction mixture contained 5 μL of DNA template, 12.5 μL of 2X PCR master mix (Thermo Scientific, United States), 6.5 μL of deionized nuclease-free water, and 1 μL of each primer. PCR conditions were 94 °C for 2 min for initial denaturation, followed by 35 cycles of 98 °C for 10 s, 58 °C for 30 s, and 68 °C for 60 s, then 68 °C for 7 min as a final extension step; this cyclic reaction was run using a 96 well Applied Biosystem, 2720 thermal cycler. The PCR products were visualized using 1% agarose gel using a UV transilluminator and a Gel Documentation System (cleaver scientific ltd UV gel documentation system, USA). *S. aureus* strain from a previous study was used as a positive control [28].

Furthermore, all the biochemically characterized *E. coli* isolates were subjected to PCR amplification for encoding of the Genus-specific primer 16S-rRNA gene using the following primer pair: 16S-rRNA-F-GCGGACGGGTGAGTAATGT and 16S-rRNA-R-TCATCCTCTCAGACCAGCTA (Table 1). PCR protocol was performed as per Teichmann et al. [29]. The reaction was performed in a total volume of 25 μL with 5 μL DNA template, 12.5 μL 2X PCR master mix (Thermo Scientific, United States), 6.5 μL deionized nuclease-free water, and 1 μL of each primer. A 96-well Applied Biosystem and a 2720 thermal cycler were used for PCR amplification. The following PCR amplification reaction was applied: initial denaturation for 5 min at 94 °C; 30 cycles of 94 °C for 15 s, 69 °C for 30 s, and 72 °C for 30 s; final extension for 7 min at 72 °C. PCR products were run on 1% agarose gel and were visualized using a UV transilluminator and a Gel Documentation System (cleaver scientific ltd UV gel documentation system, USA). The strain used as PCR positive control was supplied from a previous study [30].

### 2.5. Antimicrobial Sensitivity Test for S. aureus and E. coli Strains

All the confirmed bacterial isolates were characterized for their antimicrobial sensitivity tests on Muller Hinton agar (Oxoid, Ltd., Basingstoke, UK) by using the disc diffusion technique according to CLSI [31]. Firstly, the twenty-one *E. coli* isolates were tested for antimicrobial sensitivity against 12 different antimicrobial compounds by disc diffusion test. The following antimicrobial discs (Oxoid, Ltd., Basingstoke, UK) were used: penicillin (10 mg), cefuroxime (30 mg), cefoperazone (30 mg), amikacin (30 mg), streptomycin (15 mg), neomycin (5 mg), azithromycin (15 mg), nalidixic acid (30 mg), trimethoprim-Sulfamethazole (25 mg), gentamycin (10 mg), chloramphenicol (30 mg), and rifamycin (5 mg) (Table 2). In addition, thirty-eight *S. aureus* isolates were examined against 10 antimicrobial agents for their antimicrobial sensitivity using a disc diffusion test on Muller Hinton agar (Oxoid, Ltd., Basingstoke, UK). These antimicrobial discs (Oxoid, Ltd., Basingstoke, UK) were ampicillin (10 mg), oxacillin (15 mg), ceftazidime (30 mg), kanamycin (30 mg), streptomycin (15 mg), norfloxacin (10 mg), ciprofloxacin (5 mg), chloramphenicol (30 mg), tetracycline (30 mg), and gentamycin (10 mg) (Table 2). The results were interpreted according to CLSI [31]. Resistance to more than two antibiotics from different antimicrobial classes was recorded as MDR [32]. A multiple antibiotic resistance (MAR) index was calculated according to Krumperman [33].

### 2.6. Complete Blood Cell Analysis

Red blood cell count (RBCs), hemoglobin (Hb), hematocrit value (PCV), and total and differential leukocytic count were analyzed according to Morar et al. [34].

### 2.7. Biochemical Markers, Acute Phase Protein, and Inflammatory Cytokines Analysis

The serum activity of aspartate aminotransferase (AST, catalog No.; AS101) was estimated using kits obtained from the Randox company (Kearneysville, VA, USA). Lactate dehydrogenase (LDH, catalog No.; TK41214), total protein (Catalog No.; MD1001291), and albumin (Catalog No.; MX1001020) were assessed using kits obtained from the Spinreact company (Santa Coloma, Spain) according to the described methods of its manufacturer’s instructions.

Serum levels of acute-phase protein (APP) including haptoglobin (Catalog No.; ab137977), amyloid A (Catalog No.; ab274407), fibrinogen (Catalog No.; ab108842), ceruloplasmin (Catalog No.; ab108818), and ferritin (Catalog No.; ab108698) were estimated using kits obtained from the Abcam company (Cambridge, UK) according to the standard protocol of their specific pamphlets.

Serum tumor necrosis factor-alpha (TNF-α, Catalog No.; MBS2609886), interleukins, IL-1β, IL-6, (Catalog No.; # ESS0027, and # RBOIL6I) (Invitrogen-Thermo Fisher Co., Waltham, MA, USA) and IL-10, (Catalog No.; ab277386) (Abcam Co., Cambridge, UK) levels were estimated using specific bovine ELISA commercial kits according to the methodology of each enclosed pamphlet. All parameters were measured spectrophotometrically using a 5010 Photometer (ROBERT RIELE GmbH & Co KG, Berlin, Germany).

### 2.8. Oxidative Stress/Antioxidant Parameters

The serum levels of malondialdehyde, total antioxidant capacity (TAC) superoxide dismutase (SOD), and catalase were estimated spectrophotometrically following the illustrated approaches of Ohkawa et al. [35], Benzie and Strain [36], Nishikimi et al. [37], and Aebi [38], respectively.

### 2.9. Statistical Analysis

Data analysis was carried out using a statistical software program (SPSS for Windows, Version 21, Cary, NC, USA). The Kolmogorov–Smirnov test was selected to assess the normal distribution of the data. The assessed data were normally distributed; therefore, the means and standard mean of error (SME) for each variable were statistically analyzed and presented. The post hoc test with ANOVA (Tukey’s test) was used to assess statistical differences between the two groups. In all statistical analyses, the results were considered significant at *p* < 0.05.

## 3. Results

### 3.1. Characterization of E. coli and S. aureus among Clinical and Sub-Clinical Mastitic Milk

Out of all investigated mastitic milk samples, a total of 70% (21/30) *E. coli* isolates and 76% (38/50) *S. aureus* isolates were identified from clinical mastitis milk and sub-clinical mastitis milk, respectively, using PCR assay. All the other samples were discarded and counted as contaminated samples. A total of twenty milk samples from healthy cows were negative for bacterial culture. Due to the absence of observable clinical signs in animals infected with subclinical mastitis, a presumptive diagnosis was done based on laboratory diagnostic tests of milk samples using the CMT and SCC. Cows with positive CMT or those having SCC > 200,000 cells/mL but lacking clinical signs were considered affected with subclinical mastitis.

### 3.2. Antimicrobial Susceptibility Testing for E. coli and S. aureus among Clinical and Sub-Clinical Mastitic Milk

The results of antimicrobial susceptibility testing of *E. coli* isolates (*n* = 21) are listed in Table 2. *E. coli* isolates were resistant to penicillin, cefuroxime, cefoperazone, azithromycin, nalidixic acid, trimethoprim-sulfamethoxazole, rifamycin, and gentamycin. They displayed intermediate resistance to neomycin (61.9%), amikacin (52.4%), and chloramphenicol (47.6%). *E. coli* isolates were more sensitive to streptomycin (38.1%). Multiple drug resistance (MDR) was detected in all tested *E. coli* isolates (resistant to ≥3 antimicrobial class) and the most prevalent antimicrobial pattern was ‘’C, P, DA, STX, RA, N, NA, CXM, CEP, AZM’’ (Table 3).

Moreover, the antimicrobial susceptibility testing of *S. aureus* isolates (*n* = 38) is listed in Table 2. *S. aureus* isolates were highly resistant to ampicillin, oxacillin, and gentamycin (76.3%, 76.3%, and 73.7%), respectively, and they displayed intermediate resistance to ceftazidime (52.6%), kanamycin (47.4%), and streptomycin (50%). *S. aureus* was more sensitive to norfloxacin (34.2%), ciprofloxacin (36.8%), chloramphenicol (26.3%), and tetracycline (23.7%). Multiple drug resistance (MDR) was detected in 94.74% of the total tested isolates (resistant to ≥3 antimicrobial classes). Only two *S. aureus* strains were found to be sensitive to all antimicrobials tested. The most prevalent antimicrobial pattern was K, NOR DA, TE, OX, CIP, CAZ, AMP, and S (Table 3).

### 3.3. Hematological and Serum Biochemical Parameters of Dairy Cows with Subclinical and Clinical Mastitis

As presented in Table 4, the obtained results showed significantly lower RBCs count (*p* < 0.01), Hb (*p* < 0.001), and PCV (*p* < 0.01) values in mastitic cows compared with both subclinical mastitic and control cows. Moreover, WBCs (*p* < 0.01), and neutrophil (*p* < 0.001) counts were significantly diminished in clinical and subclinical mastitic cows compared to the controls. Significantly higher levels of AST (*p* < 0.001, *p* < 0.01), LDH (*p* < 0.001, *p* < 0.01), total protein (*p* < 0.001, *p* < 0.05), and globulin (*p* < 0.001, *p* < 0.05) in both mastitic and subclinical mastitic cows unlike that of the controls. However, the serum level of albumin (*p* < 0.05) statistically declined in mastitic cows only compared to other groups.

### 3.4. Serum Acute Phase Protein of Dairy Cows with Subclinical and Clinical Mastitis

To assess the mechanisms involved in the progression and damage of mammary gland tissue during mastitis, serum acute-phase proteins (APP) and inflammatory cytokines were estimated in our study. The haptoglobin (*p* < 0.001), fibrinogen (*p* < 0.01), and Amyloid A (*p* < 0.001, *p* < 0.01) levels were found to be highly elevated in the serum of both mastitic and subclinical mastitic cows unlike that of the controls (Figure 1). Moreover, ceruloplasmin (*p* < 0.05) levels were significantly raised in the serum of mastitic cows compared to the healthy controls (Figure 1D). No significantly valuable differences in serum ferritin levels of all the groups were identified (Figure 1E).

### 3.5. Serum Inflammatory Cytokines of Dairy Cows with Subclinical and Clinical Mastitis

TNF-α (*p* < 0.001), IL-1β (*p* < 0.001, *p* < 0.01), and IL-6 (*p* < 0.001, *p* < 0.01) were statistically increased in mastitic and subclinical mastitic cows compared to the healthy controls (Figure 2). Meanwhile, the IL-10 (*p* < 0.01) decreased only in mastitic cows relative to the remaining groups (Figure 2D).

### 3.6. Serum Antioxidant\Oxidative Stress Parameters of Dairy Cows with Subclinical and Clinical Mastitis

During the progression of mastitis, the bacterial infection caused the generation of an accentuated reactive oxygen species (ROS) and impairment of antioxidant molecules confirmed in our results by higher malondialdehyde (MDA) levels (*p* < 0.001), along with a reduction of total antioxidant capacity (TAC) (*p* < 0.05, *p* < 0.001) and catalase (*p* < 0.01, *p* < 0.001) in mastitic and subclinical mastitic cows compared to the control non-infected one (Figure 3). Superoxide dismutase (SOD) (*p* < 0.05) was statistically diminished only in mastitic cows compared to the controls (Figure 3C).

## 4. Discussion

Mastitis is a mammary gland inflammatory illness that causes huge, enormous economic losses in the dairy industry [39]. It occurs when a pathogenic microbe enters the mammary gland, usually by disrupting physical barriers like the teat canal [40]. Once the barrier is breached, a prompt and effective defense response is required to prevent the spread of pathogenic organisms and additional injury to mammary gland tissue [41]. Mastitis, in its clinical and subclinical forms, is considered one of the most devastating diseases that affect dairy herds and represents 21% of reported diseases in dairy cattle, with an annual incidence of 37% [42]. To our knowledge, there were no studies applied to clinical or subclinical mastitic cows that directly link the bacterial causative agent, and serum biochemical, antioxidant, and inflammatory markers (APP & cytokines) with mastitis risk. That is why the goal of this study was to investigate the bacterial cause and detect serum biochemical, antioxidant, and inflammatory markers in dairy cows that were linked to clinical and subclinical mastitis susceptibility.

Mastitis is a complex disease with varied etiological causes, contagious bacterial, environmental, and opportunist [43]. In our study, the bacterial agent was detected in high proportion 73.75% (59/80) of all mastitic milk samples. This confirms previous studies which mentioned bacterial mastitis as the most common etiology [44,45]. Furthermore, our findings revealed that *E. coli* and *S. aureus* were the causative agents responsible for our mastitic cases which are in accordance with previous studies that described these two microorganisms to be the most common and significant bacterial agents responsible for mastitis [46].

*E. coli* is an environmental agent capable of invading the cow’s teat through ascending infection [47]. *E. coli* was recovered from 70% (21/30) of our clinical mastitis cases. Higher results were obtained in France and Egypt; investigating more than 80% of the clinical mastitic cases were caused by *E. coli* [48,49]. In Egypt, a previous report showed a slightly low occurrence (about 31%) of clinical mastitic cases where *E. coli* was the causative agent [50]. Ahmed et al. [51] recovered a low incidence (9.1%) of *E. coli*-related mastitic cases.

*S. aureus* is a contagious microorganism that invades the udder and infects all the quarters. In our study, *S. aureus* recovered at a rate of 76% (38/50) from the subclinical cases which came following other studies in Egypt [52,53]. From Egypt, another study was found to have higher results than ours; about 91.48% of the samples were identified as *S. aureus* [54].

Antimicrobial resistance is an important hazard that is mentioned as the silent tsunami facing modern medicine. Nowadays, MDR has been a serious challenge facing scientists [55]. Examination of our isolates from farms that did not receive any previous medication the last week relived a dangerous result for MDR. Our *E. coli* mastitic strains showed extensive resistance against seven antimicrobial classes, β-lactam, cephalosporin, macrolide, quinolone, sulphonamide, lincosamide, and rifamycin. Previous reports discussed *E. coli* raised antimicrobial resistance against ampicillin, streptomycin, and sulfonamides [56,57,58]; in addition, extended-spectrum β -lactamases resistance was investigated in other reports [59,60]. These results might be attributed to the broad-spectrum antimicrobials, trimethoprim–sulfonamides, oxytetracycline, fluoroquinolones, cefquinome, and ceftiofur which are used in the farms for the systematic treatment of mastitic coliform [61,62]. Thus, *E. coli* have developed resistance to these extensively used antimicrobial classes.

In this study, the isolated *Staphylococcus aureus* strains showed slightly lower antimicrobial resistance results than the *E. coli* strain. Our study exhibited higher resistance against ampicillin which came in accordance with previous studies [53,63]. *S. aureus* produces penicillinase enzyme which makes the treatment by β-lactamase antimicrobial class more problematic. Third-generation cephalosporin was considered for treatment of the mastitis cases caused by *S. aureus*. It was revealed that about half of the strains were ceftazidime resistant which was the same as other previous studies [53]. We observed that about 73% of the strains developed resistance to gentamicin which was higher than in other studies [64,65]. Reports from China showed that antimicrobial agents such as penicillin, ampicillin, streptomycin, gentamicin, ciprofloxacin, and sulfamethoxazole-trimethoprim were usually applied for mastitis in dairy farms [66]; these antimicrobial agents act by inhibiting protein synthesis [67,68]. Resistance against penicillin, ampicillin, erythromycin, tetracycline, and clindamycin was reported in different countries [69,70,71,72]. In this study, the low resistance against tetracycline and chloramphenicol was reasonable. Since tetracycline resistance is usually acquired by horizontal gene transfer; besides, chloramphenicol is not used anymore in the veterinary field. The proportion of MDR *S. aureus* in our study was 94.74%. These results are similar to previous studies from Malaysia and Brazil [71,73].

The inappropriate use of antimicrobials in farms may be involved in the emergence or spread of antimicrobial-resistant strains; these strains could be transmitted to humans via direct contact with animals or via ingestion of infected food [74,75]. Thus, misuse led to the environmental release of the antimicrobial resistance genes and resistant bacterium. In this case, the environment and the animals will serve as an end reservoir of the antimicrobial resistance genes [76]. This limited the options used for the therapy against these agents [77]. Periodical surveillance strategies must be taken to control this dramatic development of MDR bacterial strains [78]. Nowadays, the emergence of novel strategies such as the usage of antimicrobial combination, phage therapy, and peptide therapy had been studied [79,80].

Changes in hemato-biochemical parameters and leukocyte counts can be employed as crucial indications of the animal’s physiological or pathological status (mastitis). Following our results, the mastitis-affected cows showed a significant decrease in Hb, PCV, and RBCs counts, with an increase in TLC as compared to healthy cows [81]. These findings were also consistent with a prior study that found a significant fall in RBC, Hb, and PCV values in mastitis-affected animals, resulting in anemia [82]. Earlier studies also found that mastitic cows had a higher leukocyte, granulocyte count, lymphopenia, and anemia [83]. TLC levels were recorded to be increased in affected animals, as well as monocyte, neutrophil, and eosinophil count [82]. Various immunomodulatory actions may be responsible for the considerable increase in WBC and neutrophilic counts in mastitic cows [84]. Furthermore, Abba et al. [85] attributed those neutrophils enhanced the chemotactic molecules generated by infectious pathogens, as well as other immune system components that activate neutrophil recruitment to infection sites.

Compared to healthy cows, the activities of AST and LDH exhibited a considerable increase in mastitic cows, which could be attributed to stressful conditions. Our findings were consistent with prior research [81,82,86,87]. In terms of total protein findings, our results revealed a significant increase in serum total protein and globulin with lower albumin levels in mastitic groups when compared to healthy ones. These findings agreed with Ali et al. [88] and Garba et al. [89] who found a significant increase in total protein levels with significantly lower albumin levels in cows with clinical and subclinical mastitis. Similarly, previous reports recorded high levels of globulin and total protein in mastitic cows [88,90]. The immunological response is the main cause of the decrease in albumin levels that are linked to udder infection [89]. As well, hypoalbuminemia may be interpreted as the stress that occurs during mastitis, which increases protein catabolism [91]. Increased serum globulin levels could be related to the development of antibodies in the form of gamma-globulin, which is responsible for neutralizing the invading microorganism’s effect [92]. Alternatively, Sarvesha et al. [93], and Krishnappa et al. [94] recorded the mastitic group had a significantly lower level of total proteins in mastitic in indigenous cows, crossbred cattle, and buffaloes.

To identify cows with subclinical mastitis, laboratory diagnostics is crucial. Mastitis control is primarily dependent on determining SCC and the CMT, which aim to detect the number of cells in the milk sample. The other useful diagnostic tool is microbial culture, which complements SCC and CMT [95]. However, all mentioned diagnostic methods have their limitations and therefore novel biomarkers of subclinical mastitis are highly desired. These sensitive indicators include cytokines and acute phase protein measurements, particularly haptoglobin, fibrinogen, and amyloid A which could be determined in cow serum and/or milk, and in the future may become useful in early mastitis diagnostics as well as a preventive tool. This may contribute to increase the detection of mammary gland inflammation in cows, especially in subclinical form, and consequently improve milk quality and quantity. Over the last few decades, research has revealed that acute-phase protein quantitation (APP) e.g., ceruloplasmin, fibrinogen, haptoglobin, and amyloid A, can be found in blood plasma or serum and it can be used for disease diagnosis and prognosis detection, as well as to assess the degree of inflammation [96,97]. In the current investigation, we found that ill cows had significantly higher levels of haptoglobin, fibrinogen, amyloid A, and ceruloplasmin in subclinical and clinical mastitic animals than in the healthy cows. The production of proinflammatory cytokines like TNF-α, which exacerbate the inflammatory process and stimulate neutrophil phagocytic activity, could be to blame for the increased amounts of APPs [98]. The elevated levels of APP are in accordance with those obtained by Winter et al. [99] in cows with *Listeria monocytogenes*-induced mastitis, as well as in clinical and subclinical mastitic dairy cows [100,101].

Data about subclinical and clinical mastitis demonstrate inflammatory responses to the intramammary infection driven by IL-1β, IL-6, and TNF-α. Moreover, the host defense response in mastitis is characterized by the continuation or resolution of initial inflammation [102]. Additionally, our aim of the study was to determine inflammatory and regulatory cytokines in the serum of dairy cows with subclinical and clinical mastitis and to help in the diagnosis of subclinical or clinical mastitis. Knowledge about the inflammatory and regulatory cytokines in naturally occurring mastitis is lacking as most studies focus on pathogen-induced mastitis. In naturally occurring mastitis, the study of cytokines is a potential tool for early and timely diagnosis and in prospective pathogenesis-based treatment. Our finding recorded statistically increased TNF-α, IL-1β, and IL-6 in mastitic and subclinical mastitic cows compared to the healthy controls, while the IL-10 was decreased only in mastitic cows relative to the other groups. Similarly, it has been shown that pro-inflammatory cytokines are important in fighting the original infection. Increased levels of TNF-α and IL-1β in subclinical mastitis were discovered in this study, suggesting that they play a role in the early stages of mastitis development [103,104,105]. The pro-inflammatory cytokines IL-12, IL-6, TNF-α, and IL-1β were found to be significantly elevated in lactating cows suffering from clinical mastitis [102]. Variations in these proinflammatory concentrations corresponded to the onset of illness signs. As previously reported, increased pH and significant decreases in fat, SNF, protein, and lactose were identified in the mammary gland tissue of these cows [106]. In contrast to our findings, previous research showed a non-significant rise in IL-2 and IL-6 markers in cows with subclinical mastitis [107].

All the living organisms, particularly dairy cows, produce free radicals because of their active metabolism. Oxidative stress occurs when homeostasis is disrupted, which is mostly caused by the generation and accumulation of free radicals, which can lead to mastitis in dairy cows [108]. Antioxidants protect the body from oxidative damage caused by free radicals by scavenging them directly or by inhibiting the action of oxidizing enzymes [109]. In the present study, there was a significant increase in MDA, along with a reduction of TAC, SOD, and catalase in subclinical and clinical mastitic animals compared to the control non-infected ones. The considerable increase in MDA levels, as well as the significant drop in TAC and catalase activity, such findings could imply an increase in free radical activity, which would reflect the state of oxidative stress that happens in such situations [110]. Previously similar findings have been reported by Sharma et al., [111] and Jhambh et al. [112]. Increased consumption to neutralize ROS generated by the inflamed gland could explain the decreased antioxidant enzymatic activity, indicating a weakened antioxidant defense mechanism [90]. Furthermore, these changes could be due to the high demand for high SOD, catalase, and GPx activities for elevated levels of oxidant damage caused by inflammatory reactions in the mammary gland tissue or insufficient nutrition, which has a significant impact on the level of blood lipid peroxidation and lack of energy increases blood plasma levels of MDA in clinical mastitis animals [113].

Nonetheless, this study has limitations to discuss. our baseline scenario assumed that blood parameters were only affected by the mastitis inflammation caused by the bacterial infection, while the bacterial-caused mastitis might result in a decrease in the immune system efficiency which would permit the entrance of secondary invaders or decrease cows’ food intake causing a nutritional deficiency sign. Thus, to solve those limitations a future study on broad aspects should be done where a large scale of mastitic cows in different geographical regions inside Egypt examined for different microbiological, environmental, and nutritional disorders besides, performing a correlation between the revealed data to a better understanding the linkage of mastitis inflammation and microbes. This can be accomplished with the help of new computational biology techniques or through sophisticated technology and artificial intelligence algorithms. Moreover, the study emphasizes the necessity to evaluate additional cytokines to the already studied, including Interleukin-1 alpha (IL-1α) Interleukin-4 (IL-4), Interleukin-17 (IL-17), Interleukin-13 (IL-13), cathelicidin LL37, nuclear factor kappa B (NF-κB), and transforming growth factor-beta 1 (TGF-β1). Together would explain their relationships in detailed interrelations by correlation analysis to substantiate the knowledge of the broad cytokine participating in both host immune defense and mastitis inflammation. Finally, it might be the simplest way to promote the pasteurization of milk so that humans will not get exposed to resistant bacteria.

## 5. Conclusions

In conclusion, the current findings show that dairy cow mastitis is related to significant immunological, and oxidative changes, including hematological, biochemical indicators, oxidative acute phase protein (APP), and inflammatory cytokine with the predisposing factors for mastitis resistance/susceptibility highlighted by our findings. These findings suggest that variations in these biomarkers could be utilized to diagnose such illnesses. The variable pattern of antioxidant, APP, and inflammatory cytokines in subclinical and clinical mastitic dairy cows could be a biomarker for bovine immune status which not only predicts the most susceptible risk time for disease occurrence but also builds up an effective management protocol to improve health through proper breeding and vaccination regimens.

## Figures and Tables

**Figure 1 animals-13-00892-f001:**
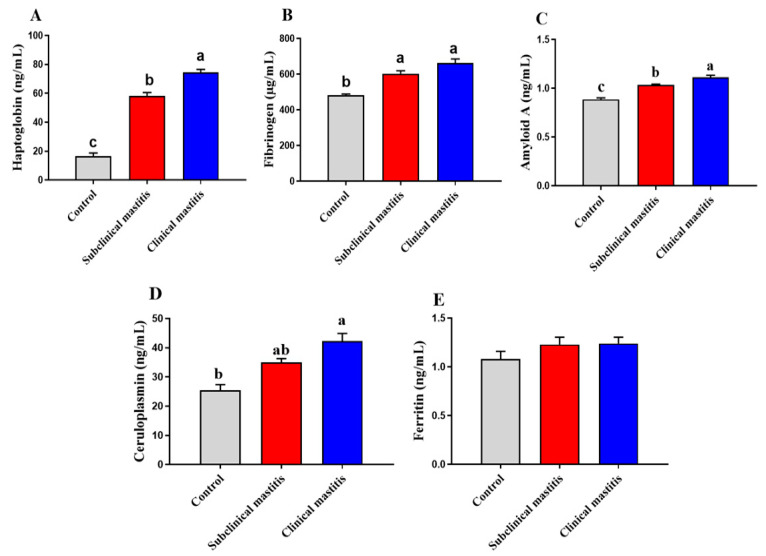
Serum Acute phase protein of dairy cows with subclinical and clinical mastitis. Data were represented as Mean ± SME. This means in with different superscripts are significantly different (*p* < 0.05). a significantly different from b and c, a for the high value, b for intermediate value between a and c, while c for the lower value. Haptoglobin (**A**), fibrinogen (**B**), Amyloid A (**C**), ceruloplasmin (**D**), and ferritin (**E**).

**Figure 2 animals-13-00892-f002:**
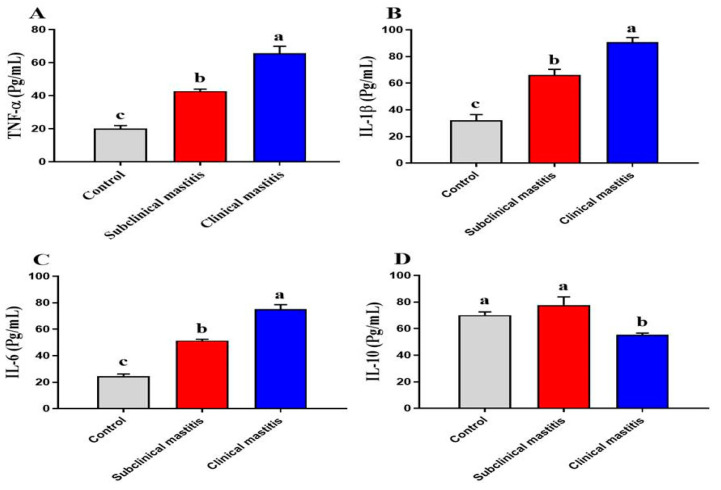
Serum inflammatory Cytokines of dairy cows with subclinical and clinical mastitis. Data were represented as Mean ± SME. This means in with different superscripts are significantly different (*p* < 0.05). a significantly different from b and c, a for the high value, b for intermediate value between a and c, while c for the lower value.TNF-α (**A**), IL-1B (**B**), IL-6 (**C**) and IL-10 (**D**).

**Figure 3 animals-13-00892-f003:**
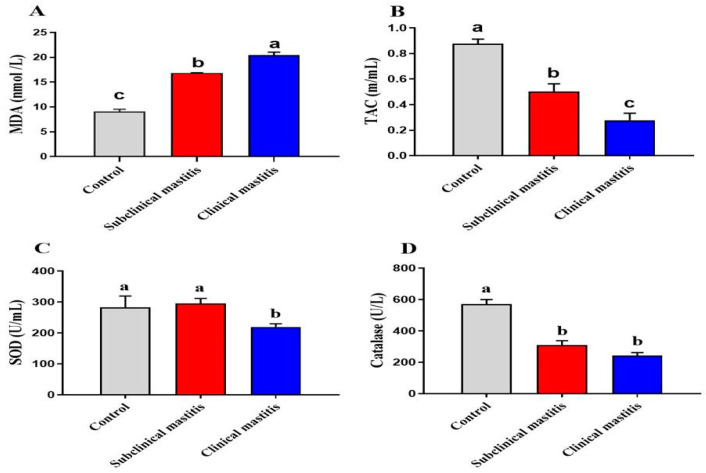
Serum antioxidant/oxidative stress parameters of dairy cows with subclinical and clinical mastitis. Data were represented as Mean ± SME. This means in with different superscripts are significantly different (*p* < 0.05). a significantly different from b and c, a for the high value, b for intermediate value between a and c, while c for the lower value. Malondialdehyde (MDA, **A**), total antioxidant capacity (TAC, **B**), Superoxide dismutase (SOD, **C**), and catalase (**D**).

**Table 1 animals-13-00892-t001:** Primers were used for amplification in this study.

Gene	Bacteria	Gene	Primer Sequences (5′ to 3′)	bp	Reference
*nuc*	*S. aureus*	*Nuc*-F	F-GTGCTGGCATATGTATGGCAATTG	660	[27]
		*Nuc*-R	R-CTGAATCAGCGTGTCTTCGCTCCAA		
*16s-rRNA*	*E. coli*	*16S rRNA*-F	F-GCGGACGGGTGAGTAATGT	200	[29]
		*16S rRNA*-R	R-TCATCCTCTCAGACCAGCTA		

**Table 2 animals-13-00892-t002:** Antimicrobial sensitivity detected in different *Escherichia coli* and *Staphylococcus aureus* isolates from clinical and sub-clinical mastitic milk.

Bacteria	Antimicrobial Agent	Family	CPD	Resistance No/%	Intermediate No (%)	Sensitive No/%
*E. coli*	Penicillin	β-lactam	10	21 (100%)	-	-
	Cefuroxime	Cephalosporin	30	21 (100%)	-	-
	Cefoperazone		30	21 (100%)	-	-
	Amikacin	Aminoglycoside	30	11 (52.4%)	8 (38.1%)	2 (9.5%)
	Streptomycin		15	8 (38.1%)	8 (38.1%)	5 (23.8%)
	Neomycin		5	13 (61.9%)	3	5 (23.8%)
	Azithromycin	Macrolide	15	21 (100%)	-	-
	Nalidixic acid	Quinolone	30	21 (100%)	-	-
	Trimethoprim- Sulfamethazole	Sulphonamide	25	21 (100%)	-	-
	Gentamycin	Lincosamide	10	21 (100%)	-	-
	Chloramphenicol	Phenicols	30	10 (47.6%)	8 (38.1%)	3 (14.29%)
	Rifamycin	Rifamycin	5	21 (100%)	-	-
*S. aureus*	Ampicillin	β-lactam	10	29 (76.3%)	5 (13.2%)	4 (10.5%)
	Oxacillin		15	29 (76.3%)	5 (13.2%)	4 (10.5%)
	Ceftazidime	Cephalosporin	30	20 (52.6%)	10 (26.3%)	8 (21.1%)
	Kanamycin	Aminoglycoside	30	18 (47.4%)	19 (50%)	1 (2.6%)
	Streptomycin		15	19 (50%)	13 (34.2%)	6 (15.8%)
	Norfloxacin	Quinolone	10	13 (34.2%)	5 (13.2%)	20 (52.6%)
	Ciprofloxacin	Fluoroquinolone	5	14 (36.8%)	14 (36.8%)	10 (26.3%)
	Gentamycin	Lincosamide	10	28 (73.7%)	10 (26.3%)	-
	Chloramphenicol	Phenicols	30	10 (26.3%)	14 (36.8%)	14 (36.8%)
	Tetracycline	Tetracycline	30	9 (23.7%)	9 (23.7%)	20 (52.6%)

**Table 3 animals-13-00892-t003:** Antimicrobial sensitivity pattern and multiple antimicrobial resistance detected in different *Escherichia coli* and *Staphylococcus aureus* isolates from clinical and sub-clinical mastitic milk.

Bacteria	Antibiotypes	Resistance Pattern	Isolates No (%)	MAR Index
*E. coli*	I	C, P, DA, STX, RA, AK, NA, CXM, CEP	1(4.8%)	0.75
	II	C, P, DA, STX, RA, NA, CXM, CEP, AZM	1(4.8%)	0.75
	III	P, DA, STX, RA, N, NA, CXM, CEP, AZM	3(14.3%)	0.75
	IV	C, P, DA, STX, RA, AK, NA, CXM, CEP, AZM	3(14.3%)	0.83
	V	C, P, DA, STX, RA, N, NA, CXM, CEP, AZM	5(23.8%)	0.83
	VI	P, S, DA, STX, RA, N, NA, CXM, CEP, AZM	1(4.8%)	0.83
	VII	P, S, DA, STX, RA, N, AK, NA, CXM, CEP, AZM	4(19.04%)	0.92
	VIII	P, S, DA, STX, RA, AK, NA, CXM, CEP, AZM	3(14.3%)	0.83
*S. aureus*	I	OX, CAZ, AMP	2(5.3%)	0.3
	II	K, DA, CAZ	4(10.5%)	0.3
	III	NOR, DA, OX, AMP	1(2.6%)	0.4
	IV	OX, CLP, CAZ, AMP	1(2.6%)	0.4
	V	NOR, OX, CAZ, AMP	3(13.2%)	0.4
	VI	K, C, DA, OX, S	5(13.2%)	0.5
	VII	C, DA, CAZ, AMP, S	1(2.6%)	0.5
	VIII	K, OX, CIP, CAZ, AMP	4(13.2%)	0.5
	IX	DA, OX, CAZ, AMP, S	4(10.5%)	0.5
	X	NOR, DA, OX, CIP, AMP	4(13.2%)	0.5

**Table 4 animals-13-00892-t004:** Hematological and serum biochemical parameters of dairy cows with subclinical and clinical mastitis.

Parameters	Control	Subclinical Mastitis	Clinical Mastitis
RBCs (10^6^/µL)	7.41 ± 0.24 ^a^	6.81 ± 0.33 ^a^	5.45 ± 0.19 ^b^
Hb (g/dL)	10.13 ± 0.15 ^a^	9.39 ± 0.10 ^a^	8.21 ± 0.07 ^b^
PCV (%)	31.51 ± 0.70 ^a^	29.02 ± 0.71 ^a^	26.38 ± 0.13 ^b^
WBCs (10^3^/µL)	10.32 ± 0.37 ^c^	13.44 ± 0.09 ^b^	18.05 ± 0.71 ^a^
Lymphocyte (10^3^/µL)	7.56 ± 0.16	8.07 ± 0.05	8.34 ± 0.66
Neutrophil (10^3^/µL)	2.32 ± 0.29 ^c^	4.46 ± 0.06 ^b^	9.08 ± 0.36 ^a^
Monocyte (10^3^/µL)	0.44 ± 0.18	0.91 ± 0.05	0.64 ± 0.26
AST (U/L)	46.17 ± 2.24 ^c^	66.45 ± 1.22 ^b^	94.57 ± 2.08 ^a^
LDH (U/L)	261.21 ± 14.88 ^c^	479.77 ± 9.56 ^b^	766.23 ± 10.81 ^a^
T. protein (g/dL)	7.43 ± 0.09 ^c^	8.37 ± 0.08 ^b^	9.02 ± 0.24 ^a^
Albumin (g/dL)	3.47 ± 0.15 ^a^	3.20 ± 0.36 ^a^	2.48 ± 0.11 ^b^
Globulin (g/dL)	3.97 ± 0.22 ^c^	5.17 ± 0.68 ^b^	6.54 ± 0.31 ^a^

Data were represented as Mean ± SME. Means in the same row with different superscripts is significantly different (*p* < 0.05). AST, Aspartate aminotransferase; LDH, lactate dehydrogenase.

## Data Availability

All the data generated or analyzed during this study are available from the corresponding author upon reasonable request.

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
