# Peer review of "Immunological and Oxidative Biomarkers in Bovine Serum from Healthy, Clinical, and Sub-Clinical Mastitis Caused by Escherichia coli and Staphylococcus aureus Infection"

_animals, 2023, doi:10.3390/ani13050892_

Round 1

Reviewer 1 Report

1. There are many pathogenic bacteria causing clinical and subclinical mastitis. The author only confirmed that the samples were infected by E. coli and S. aureus. However, it was not confirmed that the samples used in the experiment were not infected by other pathogens. Therefore, the experimental materials are not credible for the current topic.

2. Previously, many studies had confirmed that compared with healthy controls, serum inflammatory cytokines, such as TNF- α, IL-1 β and IL-6, were significantly increased in mastitic and subclinical mastitic cows compared to the healthy controls. Therefore, there is no novelty in the detection of inflammatory factors.

3. Many moleculars detected by the author, such as inflammatory factors and acute phase proteins (APP), are differentially expressed in many inflammatory diseases including mastitis. In addition, the author did not rule out whether the experimental animals had other diseases. Therefore,  it is not rigorous that APP  and cytokines, along with antioxidant markers are used as  early indicators of E. coli or S. aureus type mastitis.

Reviewer 2 Report

Dear authors:

The topic of your paper contain very important information about of one of the bovine dairy health problems. Without a doubt, all data about biomarkers related with mastitis cases could be specific information to clinician responsables of the dairy herd health.

However, the paper has significant opportunities for improvement by means a better description,  improve the writing and verify of data tables.

Reviewer 3 Report

The manuscript titled “Immunological and oxidative biomarkers in bovine milk from healthy, clinical, and sub-clinical mastitis milk during E. coli and S. aureus infection” compares various biomarkers in serum, and whole blood between cattle diagnosed with mastitis, sub-clinical mastitis and healthy cows as well as assesses the antimicrobial resistance status of the bacteria cultured from milk samples of these cows. While this information can be useful to veterinarians and producers, there are a few issues that need to be addressed before publication.

It is unclear to me what the purpose of the antimicrobial resistance data is. If the purpose was to assess AMR as a public health risk, then the question arises of whether these bacteria, if shed in the milk and consumed by people, aren’t killed during pasteurization. Is there a large market for raw milk consumption or is there any other way the public could be confronted with these bacteria?

If the goal was to assess whether veterinary drugs would be effective in the treatment of mastitis in cattle, then the use of human breakpoints, which I assume was the case from the cited reference, is questionable as there are veterinary breakpoints for at least some of the bacteria and antimicrobial combinations tested. Further, if veterinary AMR is the problem addressed here, then the question arises of whether all those antimicrobials tested would be used for the treatment or prevention of mastitis. In any case, the choice of breakpoints and antimicrobials needs to be discussed.

It would also be important to state whether any of the cows were recently treated with antimicrobials or when these treatments took place relative to sampling. It is known that recent antimicrobial treatment drastically increases the chances of finding resistant bacteria to the same or other antimicrobials so this information is crucial for interpretation of the findings.

The statistical analysis uses t-tests to compare normally distributed variables between groups. This approach does not take into account any adjustments for multiple comparisons between groups. ANOVA with post-hoc adjustments, such as Tukey or Scheffe (if comparing groups to a control only) would be more appropriate.

There are also a number of grammatical and stylistic issues with how the manuscript was written. Details are listed below:

Abstract:

There is nothing about antimicrobial resistance testing in the abstract, which seems to be a big part of the investigation. Please add a sentence or two about the idea and results referring to AMR.

Introduction

Line 49: this sentence needs rewriting, I’m not sure what “even worker to consumers” means in this context.

Line 52: antimicrobial residues do not lead to an increase in SCC in milk. It is mastitis that leads to an increased SCC. Also, there shouldn’t be antimicrobial residues in milk if withdrawal periods are followed.

Line 55: we always use clinical signs when speaking of animals because they themselves cannot describe their symptoms.

Line 57: I suggest “lack” instead of “disappearance” because disappearance would suggest that the signs were there first but then disappeared. Also, it ‘takes” not ‘talks” days or weeks.

Line 70 The sentence starting “Similarly, expression of…” is not a complete sentence.

Objective and hypothesis

From your objective, it sounds like you want to be able to detect which bacterial species affects the udder based on the biochemical markers. Do you think that is realistic? There should also be a statement about the objectives for the AMR testing as well as what your hypotheses were for these objectives.

Materials and Methods:

Animal population: Are the cows from one or multiple dairy farms – this is not clear from the description.

Line 84: what does “regularly” using milking machines mean? Are they using them for every milking or not?

Line 93 and following: The CMT has multiple scores; can you describe which scores you considered positive?

Sample Collection

Line 110 and ff: this paragraph is written as if the procedure will be conducted in the future. Why?

Line 129 - 131: This section is written as a set of instructions. It should be rewritten as what was performed.

Line 135: plain instead of plane.

DNA extraction

Line 140: I would change to “DNA was extracted from all suspected isolates by the boiling method” for clarity.

PCR

Line 164: “deionized water” instead of “deionized free water

CBC

Line 196: please state the name of the author instead of just the number of the reference; here and throughout the manuscript.

biochemical markers etc.

Line 204: standard instead of stander

Instead of the catalog no. please state the name of the kit, the manufacturer’s name, city and country in a consistent fashion for all such products (or according to instructions to authors for commercial products used)

Statistical analysis

as stated above, ANOVA should be used instead of multiple t-tests to compare 3 groups, with appropriate post-hoc testing.

Results:

Table 2: What is CPD? Change NO to No

Line 229: So all clinical samples were E. coli and all subclinical were S. aureus?

antimicrobial susceptibility testing

Can you list the MIC used for each antimicrobial tested? Also, please state whether these are human or veterinary breakpoints.

Line 253: please italicize S. aureus

Hematological and serum biochemical parameters

Line 265: I suppose you mean Table 4

Line 281: this is not a complete sentence, please rewrite

Line 292: please make sure all abbreviations are written out at first mention, such as ROS or MDA.

Figures 1 – 3: you should use the SE instead of SD for uncertainty around measurements for these figures. Please read Altman and Bland: Standard deviations and standard errors, BMJ, 2005 Vol  331, p 903.

Table 4: please also include reference ranges for these markers

Discussion

line 342: this should probably be incidence instead of prevalence. Prevalence would imply that 37% of cows are suffering from mastitis at any given time.

Line 349: detected instead of investigated

Line 355: E. coli arises from fecal contamination of the environment

Line 361: S. aureus infects all quarters not just the teats

Line 365 and ff: Please discuss any recent treatments with antimicrobials in the tested cows.

Line 393: This statement is too strong. There are a lot of studies that show that AMR arises even in the absence of any antimicrobial pressure. At the very least you should rephrase this to say that antimicrobial use is a contributor rather than the cause of AMR in bacteria.

Line 434: the current investigation found that ill ewes…?

Conclusion:

How do you suggest the biomarkers could be implemented on farm for diagnosis of mastitis? What are the economics and the turnaround time for using these diagnostics? I’m not saying they are useless as diagnostics, but you need to expand in your discussion about how exactly they could help farmers or veterinarians in the timely and correct diagnosis of mastitis in dairy cows, which is truly a huge problem in the dairy industry.

Round 2

Reviewer 1 Report

All the issues mentioned in the response for my comment are the main points supporting the experimental design and conclusion. Therefore, it is not enough for authors to reply only in response text. The author must supplement corresponding contents in the text, including corresponding test results, pictures and data. Otherwise, I personally think this article cannot be published unless it is well supplemented.

Author Response

Comments and Suggestions for Authors

All the issues mentioned in the response for my comment are the main points supporting the experimental design and conclusion. Therefore, it is not enough for authors to reply only in response text. The author must supplement corresponding contents in the text, including corresponding test results, pictures and data. Otherwise, I personally think this article cannot be published unless it is well supplemented.

Thanks, dear reviewer for all your valuable comments we tried hard to meet your expectations, sending the manuscript again to a postgraduate student and professors at NCSU to improve the quality of the English language of the manuscript. 

1- Answer: Thanks, for your valuable comments. We totally understand the reviewer's point of view; we improved the discussion section and added our response clearly in the text as mentioned here. As well as, we have re-analyzed our data using a post hoc test with ANOVA (Tukey's test) in the previous revision. Means and standard Mean of error (SME) were recorded to illustrate the data. This results in improving all the figures.

Reviewer 3 Report

Thank you for your revised manuscript. A lot of my concerns have been addressed, however, I still think your manuscript can be improved before publication.

Abstract

Line 36: “...compared to controls.”

The following sentence is not complete. Same for the sentence starting in line 39.

Line 40: I suggest “the findings suggested potential public health hazards due to ...”

Line 60: I’m still not sure what this sentence means. How are mastitis pathogens transmitted to consumers “through workers”. Please explain.

Line 61: this sentence is awkward. Maybe something like: The use of antimicrobial agents on catttle farms is often useful but excessive use to control or prevent mastitic infections contributes to antimicrobial resistance  development globally.

Line 66: I was taught in vet school that the word symptom only applies to human medicine, hence I recommended the term clinical signs.

Line 72: “The number of somatic cells...”

Line 95: “The study was performed on animals that had not received any medication in the week before sampling.”

Line 122: Please also indicate which scores you considered positive

Line 130: I’m sure you didn’t wipe the entire udder with alcohol? I’m not sure how you collect milk with a syringe?

Line 138: blood counting – you probably mean complete blood count?

Line 168: please state the author’s name

Line 216: delete “were”

Line 235: i don’t think index should be changed to indexes here

Line 248: I’m not sure what these catalog numbers are. Do they refer to kits to test activities for these enzymes or proteins? If yes, you should state the name of the manufacturer as elsewhere, not the catalog no. Also, the place where the kits were purchased (MyBiosource) is not important, only who made the kit.

Line 292: “All other samples...”

Paragraph 3.2: I don’t understand the capitalization of the different antimicrobials – can you make it consistent?

Line 306: this is not a complete sentence

Line 311: do you mean table 2 here?

Line 339: please use “controls” instead of “control one” – you had more than one control cow, right? Here and throughout the manuscript.

Line 316: I don’t understand what this sentence means: do you mean two isolates? What do you mean by sensitive? Sensitive to all antimicrobials tested?

Line 463: row instead of raw

Discussion

Line 622: You should point out that S. aureus was recovered from subclinical mastitis cases only.

Line 628: So they could have received antimicrobials 1 week prior to the study? That does not seem like a long wash-out period for AMR to go away based on treatment. You should comment on the fact that prior treatment could have an effect on what you found if you only have treatment records going back one week.

Line 651: Are the drugs you are mentioning typically used for mastitis treatment in Egypt? They could also be used for other infections, e.g. metritis where E. coli is also involved.

Line 672: “Inappropriate misuse” is a double negative – either use “the inappropriate use” or “the misuse”. Also, again, this statement is too strong. We still don’t understand a lot about AMR so you are oversimplifying.

You should also mention that the limitations of your findings – how will these parameters be used on the farm to identify cows with mastitis? There may be applications for these parameters in the future or through sophisticated technology and artificial intelligence algorithms – maybe. However, you do not talk about the specificity of these findings – how many other diseases cause the same type of changes? Also, is the problem with AMR that too many cows are treated or the wrong cows or with the wrong antibiotic? You should try and tie your two approaches together – how can the blood parameters help with AMR? Finally, for now, it might be simplest to promote pasteurization of milk so that humans will not get exposed to the resistant bacteria. I understand this is not popular in Egypt, but it certainly merits mention as a simple solution to the problem you are describing.

Author Response

Reviewer 3

Thank you for your revised manuscript. A lot of my concerns have been addressed, however, I still think your manuscript can be improved before publication.

Thanks, dear reviewer for your kind reply. We are trying to meet your expectation and stick with all your valuable comments in order to improve the quality of the manuscript, sending the manuscript again to a postgraduate student and professors at NCSU to improve the quality of English language of the manuscript. 

Abstract

Line 36: “...compared to controls.”

Answer. Thanks for your valuable comments. The sentence was edited and corrected as mentioned.

The following sentence is not complete. Same for the sentence starting in line 39.

Answer. Thanks for your valuable comments. The sentence was edited and corrected as mentioned.

Line 40: I suggest “the findings suggested potential public health hazards due to ...”

Answer. Thanks for your valuable comments. The sentence was edited and corrected as mentioned.

Line 60: I’m still not sure what this sentence means. How are mastitis pathogens transmitted to consumers “through workers”. Please explain.

Answer. Thanks for your valuable comments. I added below some texts and references discussing the probability of transmission of infection through workers. As workers got infected by diseased animals then the workers transmit the infection through their surroundings (family, environment, or foods).

The epidemiology of these infections is quite complex, also considering the possibility of strain exchanges between humans and animals and vice versa (Parisi et al., 2016). In recent years, the continuous detection of MRSA in food of animal origin has raised concerns about its transmission to humans via handling and/or consumption of food (EFSA, 2009; Larsen et al., 2016). during subclinical mastitis, these organisms could be shed in milk without organoleptic alterations, allowing them to enter and spread through the food chain.

Christiane et al. (2009) stated individuals who have contact with animals are more likely to be colonized with S. aureus and they also observed antibiotic usage prior to sampling has no risk with respect to colonization. It is very difficult to separate the potential effects of confounding risk factors for disease occurrence from the effect of management changes that have been adopted by a dairy farm in the production systems.

Normanno Giovanni, Spinelli Elisa, Caruso Marta, Fraccalvieri Rosa, Capozzi Loredana, Barlaam Alessandra, Parisi Antonio, Occurrence and characteristics of methicillin-resistant Staphylococcus aureus (MRSA) in buffalo bulk tank milk and the farm workers in Italy, Food Microbiology, Volume 91, 2020, 103509, ISSN 0740-0020, https://doi.org/10.1016/j.fm.2020.103509.

Marilyn C Roberts, Gemina Garland-Lewis, Sally Trufan, Scott J Meschke, Heather Fowler, Ryan C Shean, Alexander L Greninger, Peter M Rabinowitz, Distribution of Staphylococcus species in dairy cows, workers and shared farm environments, FEMS Microbiology Letters, Volume 365, Issue 15, August 2018, fny146, https://doi.org/10.1093/femsle/fny146

EWEN C. D. TODD, JUDY D. GREIG, CHARLES A. BARTLESON, BARRY S. MICHAELS; Outbreaks Where Food Workers Have Been Implicated in the Spread of Foodborne Disease. Part 6. Transmission and Survival of Pathogens in the Food Processing and Preparation Environment. J Food Prot 1 January 2009; 72 (1): 202–219. doi: https://doi.org/10.4315/0362-028X-72.1.202

Line 61: this sentence is awkward. Maybe something like: The use of antimicrobial agents on catttle farms is often useful but excessive use to control or prevent mastitic infections contributes to antimicrobial resistance  development globally.

Answer. Thanks for your valuable comments. The sentence was edited and corrected as mentioned.

Line 66: I was taught in vet school that the word symptom only applies to human medicine, hence I recommended the term clinical signs.

Answer. Thanks for your valuable comments. The sentence was edited and corrected as mentioned.

Line 72: “The number of somatic cells...”

Answer. Thanks for your valuable comments. The sentence was edited and corrected as mentioned.

Line 95: “The study was performed on animals that had not received any medication in the week before sampling.”

Answer. Thanks for your valuable comments. The sentence was edited and corrected as mentioned.

Line 122: Please also indicate which scores you considered positive we didn’t test animals with observable clinical signs for CMS

Answer. Thanks for your valuable comments. For the animals, for observable clinical signs (clinical mastitis) we didn’t test them for CMS and the farm as well.

Line 130: I’m sure you didn’t wipe the entire udder with alcohol? I’m not sure how you collect milk with a syringe? vials

Answer. Thanks for your valuable comments. The sentence was edited and corrected by removing the udder and replaced by teats and the samples were collected in one used capped vial.

Line 138: blood counting – you probably mean complete blood count?

Answer. Thanks for your valuable comments. The sentence was edited and corrected as mentioned.

Line 168: please state the author’s name

Answer. Thanks for your valuable comments. The author's name was added.

Line 216: delete “were”

Answer. Thanks for your valuable comments. The word was deleted.

Line 235: i don’t think index should be changed to indexes here

Answer. Thanks for your valuable comments. The word was edited. 

Line 248: I’m not sure what these catalog numbers are. Do they refer to kits to test activities for these enzymes or proteins? If yes, you should state the name of the manufacturer as elsewhere, not the catalog no. Also, the place where the kits were purchased (MyBiosource) is not important, only who made the kit.

Answer. Thanks for your valuable comments. Yes, it is the catalog number; we have added the company and the state of each kit (The serum activity of aspartate aminotransferase (AST, catalog No.; AS101) was estimated using kits obtained from Randox company (Virginia, USA), as well as lactate dehydrogenase (LDH, catalog No.; TK41214), total protein (Catalog No.; MD1001291), and albumin (Catalog No.; MX1001020) were assessed using kits obtained from Spinreact company (Santa Coloma, Spain) according to the described methods of its manufacturer's instructions).

Moreover, All acute-phase protein were estimated using kits obtained from Abcam Company (Cambridge, UK); we have corrected this mistake; (Serum levels of acute-phase protein (APP) including haptoglobin (Catalog No.; ab137977), Amyloid A (Catalog No.; ab274407), fibrinogen (Catalog No.; ab108842), ceruloplasmin (Catalog No.; ab108818), and ferritin (Catalog No.; ab108698) were estimated using kits obtained from Abcam company (Cambridge, UK) according to the stander standard protocol of their specific pamphlets)

Line 292: “All other samples...”

Answer. Thanks for your valuable comments. The sentence was edited and corrected as mentioned.

Paragraph 3.2: I don’t understand the capitalization of the different antimicrobials – can you make it consistent?

Answer. Thanks for your valuable comments. The sentence was edited and corrected as mentioned.

Line 306: this is not a complete sentence

Answer. Thanks for your valuable comments. The sentence was checked and edited.

Line 311: do you mean table 2 here?

Answer. Thanks for your valuable comments. The number was checked and edited and corrected as mentioned.

Line 339: please use “controls” instead of “control one” – you had more than one control cow, right? Here and throughout the manuscript.

Answer. Thanks for your valuable comments. The sentence was edited and corrected as mentioned.

Line 316: I don’t understand what this sentence means: do you mean two isolates? What do you mean by sensitive? Sensitive to all antimicrobials tested?

Answer. Thanks for your valuable comments. The sentence was edited by adding Sensitive to all antimicrobials tested Line 463: row instead of raw

Discussion

Line 622: You should point out that S. aureus was recovered from subclinical mastitis cases only.

Answer. Thanks for your valuable comments. The sentence was edited and corrected as mentioned.

Line 628: So they could have received antimicrobials 1 week prior to the study? That does not seem like a long wash-out period for AMR to go away based on treatment. You should comment on the fact that prior treatment could have an effect on what you found if you only have treatment records going back one week.

Answer. Thanks for your valuable comments. The farms we visited consider the hygiene of their animals, thus their animals don’t receive medication if there is no need or even as growth promotors. However, before visiting the farms for taking the samples we asked about the medication history, and they ensured that the animals didn’t receive any medication during the last periods so to be sure we just mentioned they didn’t receive any medication for a week.

Line 651: Are the drugs you are mentioning typically used for mastitis treatment in Egypt? They could also be used for other infections, e.g. metritis where E. coli is also involved.

Answer. Thanks for your valuable comments. We chose the antibiotics based on selecting broad aspects of antibiotics representing different antimicrobial classes

Line 672: “Inappropriate misuse” is a double negative – either use “the inappropriate use” or “the misuse”. Also, again, this statement is too strong. We still don’t understand a lot about AMR so you are oversimplifying.

Answer. Thanks for your valuable comments. The sentence was edited and corrected.

You should also mention the limitations of your findings – how will these parameters be used on the farm to identify cows with mastitis? There may be applications for these parameters in the future or through sophisticated technology and artificial intelligence algorithms – maybe. However, you do not talk about the specificity of these findings – how many other diseases cause the same type of changes? Also, is the problem with AMR that too many cows are treated or the wrong cows or with the wrong antibiotic? You should try and tie your two approaches together – how can the blood parameters help with AMR? Finally, for now, it might be simplest to promote pasteurization of milk so that humans will not get exposed to the resistant bacteria. I understand this is not popular in Egypt, but it certainly merits mention as a simple solution to the problem you are describing.

Answer. Thanks for your valuable comments. A limitation was added.

Nonetheless, this study has limitations to discuss. (1) our baseline scenario assumed that blood parameters were only affected by the mastitis inflammation caused by the bacterial infection while the bacterial-caused mastitis might result in a decrease in the immune system efficiency which would permit the entrance of secondary invaders or decrease cows' food intake causing a nutritional deficiency sign. Thus, to solve those limitations a future study on broad aspects should be done where a large scale of mastitic cows in different geographical regions inside Egypt examined for different microbiological, environmental, and nutritional disorders besides, performing a correlation between the revealed data to a better understanding the linkage of mastitis inflammation and microbes. This can be accomplished with the help of new computational biology techniques or through sophisticated technology and artificial intelligence algorithms. Moreover, the study emphasizes the necessity to evaluate additional cytokines to the already studied; including IL-1α, IL-4, IL-17, IL-13, cathelicidin LL37, nuclear factor kappa B, and transforming growth factor beta 1, all together would explain their relationships in detailed interrelations by correlation analysis to substantiate the knowledge on the broad cytokine participating in both host immune defense and mastitis inflammation. Finally, it might be the simplest way to promote the pasteurization of milk so that humans will not get exposed to resistant bacteria.
